



# Statistical post-processing of precipitation forecasts using circulation classifications and spatiotemporal deep neural networks

Tuantuan Zhang[1], Zhongmin Liang[1], Wentao Li[1,2], Jun Wang[1], Yiming Hu[1], Binquan Li[1]

[1]College of Hydrology and Water Resources, Hohai University, Nanjing 210098, China

[2]CMA-HHU Joint Laboratory for HydroMeteorological Studies, Nanjing, Jiangsu,

*Correspondence to*: Zhongmin Liang(zmliang@hhu.edu.cn)

**Abstract.** Statistical post-processing techniques are widely used to reduce systematic biases and quantify forecast uncertainty in numerical weather prediction (NWP). In this study, we propose a method to correct the raw daily forecast precipitation by combining large-scale circulation patterns with local spatiotemporal information such as topography and

meteorological factors. Particularly, we first use the self-organizing map (SOM) model to classify large-scale circulation patterns for each season, then build the convolutional neural network (CNN) model to extract spatial information (e.g., elevation, specific humidity, and mean sea level pressure) and long short-term memory network (LSTM) model to extract time series (e.g., t, t-1, t-2), and finally correct local precipitation for each circulation pattern separately. Furthermore, the proposed method (SOM-CNN-LSTM) is compared with other benchmark methods (i.e., CNN, LSTM, and CNN-LSTM) in

the Huaihe River basin with a lead time of 15 days from 2007 to 2021. The results show that the proposed SOM-CNN-LSTM post-processing method outperforms other benchmark methods for all lead times and each season with the largest correlation coefficient improvement (32.30%) and root mean square error reduction (26.58%). Moreover, the proposed method can effectively capture the westward and northward movement of the western Pacific subtropical high (WPSH), which impacts the basin's summer rain. The results illustrate that incorporating large-scale circulation patterns with local

spatiotemporal information is a feasible and effective post-processing method to improve forecasting skills, which would benefit hydrological forecasts and other applications.

## 1 Introduction

Precipitation is an important component of the global water cycle and a fundamental driver of surface hydrological processes, such as flood and drought (Xu et al. 2022). In particular, floods generated by heavy precipitation can cause a wide range of

costly, disruptive, and dangerous consequences (Herman and Schumacher, 2018). Accurate and reliable precipitation forecasts are vital for flood disaster warnings and water resource management. As the dominant way of precipitation forecasting (Bauer et al. 2015), numerical weather prediction (NWP) can provide forecast information within two weeks and the forecast skills continue to improve by about one day per decade.





However, due to the chaotic nature of the model dynamics and multisource deficiencies of the NWP models, such as
initial condition, boundary condition errors, and model structural errors, raw forecasts usually exhibit systematic and random
errors that are rapidly magnified in time (Vannitsem et al. 2021; Gneiting and Raftery, 2005). In order to reduce systematic
biases and quantify forecast uncertainty, statistical post-processing techniques are often employed, which can be divided into
parametric and nonparametric methods statistically (Li et al. 2022b). Classical parametric methods based on distribution
assumptions include bayesian model averaging (BMA) (Raftery et al. 2005), ensemble model output statistics (EMOS)
(Scheuerer and Hamill, 2015), and bayesian joint probability (BJP) (Shrestha et al. 2015). Nonparametric methods contain
quantile regression (Bremnes, 2004), ensemble copula coupling (ECC) (Schefzik et al. 2013), and the schaake shuffle (SSH)
(Clark et al. 2004), and the latter two methods can consider space-time variability and reestablish the dependence structure.

Besides the above traditional methods, machine learning (ML) methods, with the advantages of strong self-learning ability
and dealing with nonlinear problems, have been used in statistical post-processing in recent years (Ghazvinian et al. 2021;
Zhang and Ye, 2021; Peng et al. 2020). Especially, these methods can calibrate the model by using a variety of predictor-
related characteristics as input variables. Furthermore, the recent developments in deep learning, especially the convolutional
neural networks (CNN), have enabled it to be applied in the meteorological domain by taking into account high-dimensional
structured spatial data (Pan et al. 2019; Veldkamp et al. 2021). For example, Li et al. (2022b) adopted the CNN model to
correct raw forecast precipitation by considering multi-spatial information such as temperature, total column water, mean sea
level pressure, and specific humidity.

Precipitation is not only influenced by large-scale circulation systems (e.g., the western Pacific subtropical high, the South
Asian High) but also by local topography and meteorological elements (e.g., elevation, specific humidity, and mean sea level
pressure), their interaction together determines the location, intensity, and duration of precipitation (Liu et al. 2016; Ning et
al. 2017). For instance, the July 2021 extraordinary rainfall events in Henan ("21·7") happened under an abnormally strong
northerly western Pacific subtropical high, and the topographic blocking effect from the Funiu Mountain and Taihang
Mountains (Zhang et al. 2022; Zhang et al. 2021). In addition, the meteorological information from a few days ago will have
an impact on the precipitation. However, the aforementioned post-processing methods (e.g., BMA, EMOS, and BJP) usually
do not effectively incorporate large-scale circulation patterns with local spatiotemporal information. The self-organizing map
(SOM) is a nonlinear cluster technique, which has been widely used to identify large-scale circulation patterns and determine
their possible effects on local-scale precipitation and temperature (Horton et al. 2015; Loikith et al. 2017). The CNN-LSTM
model can effectively combine the advantages of CNN in processing spatial information and LSTM in processing time series,
and has been applied in precipitation fusion (Wu et al. 2020), soil moisture prediction (Li et al. 2021), and flood prediction
(Chen et al. 2022). In this study, we aim to combine the SOM technique and CNN-LSTM model to correct the raw forecast
precipitation and thus propose the SOM-CNN-LSTM post-processing method. First, considering the influence of large-scale
circulation on local precipitation, we use the SOM model to classify large-scale circulation patterns in the target basin.
Second, we build the CNN-LSTM model to extract spatiotemporal information (e.g., elevation, specific humidity, and mean
sea level pressure) and correct local precipitation for each circulation pattern separately.



This study mainly focuses on the following three questions: (1) The effectiveness of using the SOM model for large-scale circulation classification. (2) Will building a SOM-CNN-LSTM model separately for each circulation pattern improve the

quality and usefulness of precipitation forecasts? (3) Will using the CNN-LSTM model to extract spatiotemporal information enhance precipitation forecast skills?

The rest of this paper is organized as follows. Section 2 describes the study area and datasets. Section 3 describes the details of the SOM model and the CNN-LSTM model. Sections 4 and 5 present the results and discussion, respectively. The conclusion of the current research is drawn in the last section.

## 2 Study area and datasets

### 2.1 Study area

In this study, we choose the Huaihe River basin as the research area. The Huaihe River basin (30°55'~36°20'N, 111°55'~121°20'E) is located in the east of China and has an area of 270,000 km², including two major water systems: Huaihe River and Yishusi River (Fig. 1). Due to the effect of complex circulation systems, the precipitation has significant

inter-annual differences in this area, and the annual distribution is extremely uneven. The rainfall in the flood season (June to September) accounts for about 50-75% of the annual precipitation(700mm-1600mm). The Huaihe River basin is located at the boundary of the north and south climate, and the monsoon climate is very prone to heavy rains or plum rains, which can cause floods. Therefore, accurate precipitation forecast is critical to decision-making and disaster prevention (Liu et al. 2013).

### 2.2 Datasets

In this study, we choose the CN05.1 dataset as the standard precipitation data. The CN05.1 dataset is constructed based on over 2400 observing stations following the 'anomaly approach', which is a spatial resolution of 0.25° × 0.25° (Wu and Gao, 2013). we select the daily precipitation from 2007 to 2021 for calibrating and validating the forecast dataset.

TIGGE (THORPEX Interactive Grand Global Ensemble) database collects ensemble forecasts generated by thirteen numerical weather prediction (NWP) centers (Bougeault et al. 2010), such as European Centre for Medium-Range Weather

Forecasts (ECMWF), National Centers for Environmental Prediction (NCEP) and China Meteorological Administration (CMA). ECMWF consists of one control forecast and 50 perturbed forecasts generated by perturbed initial conditions, with a spatial resolution of 0.5° × 0.5°. Previous studies have compared the performance of different TIGGE products and suggested that ECMWF outperforms other products in most cases (Hamill, 2012; Huang and Luo,2017; Li et al. 2022a). Therefore, in this study, we use the ECMWF dataset and download a 51-member ensemble forecast of precipitation for the

lead time of 15 days initialized at 00UTC every day. We choose meteorological factors and topography as predictors. Meteorological factors include mean sea level pressure, U and V components of wind at 500/850/1000 hPa, 10 m U and V wind components, and specific humidity at 500/850/1000 hPa. Among them, humidity can reflect the water vapor availability, sea level pressure and wind components can reflect the moisture transport (Li et al. 2020). We also use elevation



to represent the topography, which is downloaded from the Geospatial Data Cloud of China and further extracted by ArcGIS

software. Considering that the ensemble means usually contains most of the information in the ensemble forecast, we only use the 51-member mean for all predictors. The above predictors are resampled to 0.25° with the bilinear interpolation technique. Besides, 500 hPa geopotential height anomalies with a lead time of 15 days are selected to describe the large-scale circulation patterns. Forecast precipitation, meteorological factors, and 500 hPa geopotential height are from the forecast dataset of ECMWF and can be downloaded from the following website: https://apps.ecmwf.int/datasets/data/tigge.

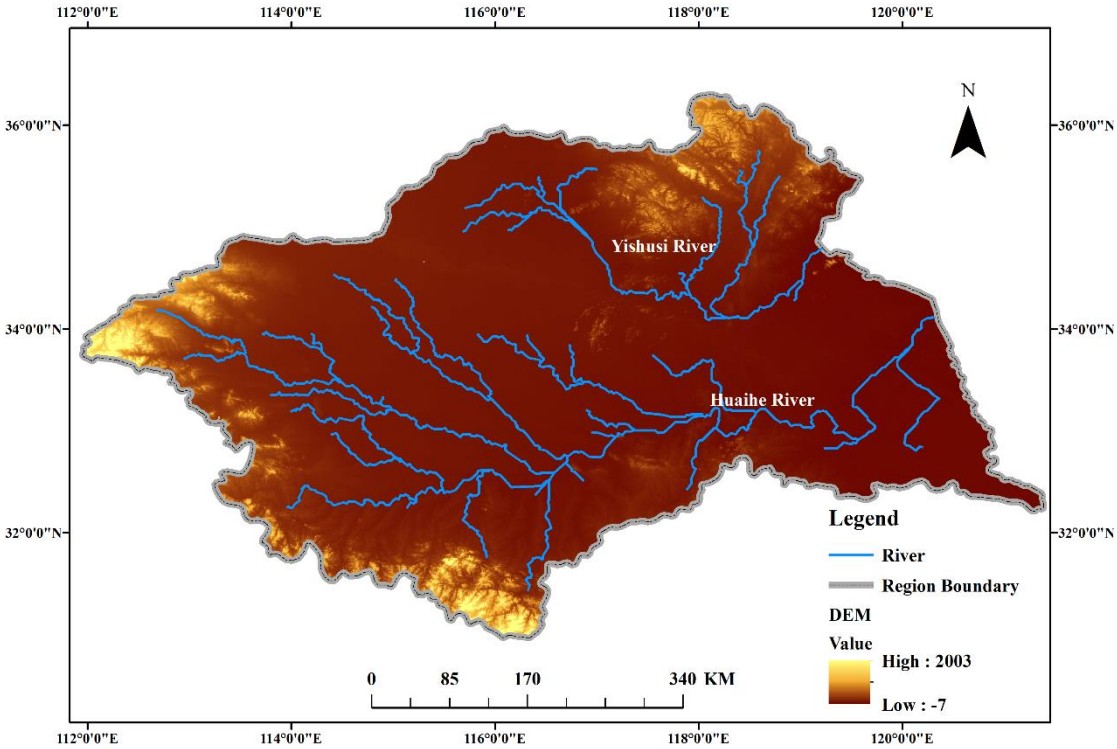


**Figure 1** Overview of the topography and rivers in the Huaihe River basin

## 3 Study area and datasets

Fig. 2 presents a flowchart of the proposed SOM-CNN-LSTM post-processing methodology for ECMWF forecasting precipitation. First, we adopt the SOM model to get the large-scale circulation patterns over the Huaihe River basin for each

lead time. Second, at each lead time, we build a CNN-LSTM model for each circulation pattern separately to correct local precipitation. Due to the significant seasonal difference in ECMWF raw forecast precipitation skills, we build statistical postprocessing models for each season separately. Details about the SOM and CNN-LSTM models will be presented in Sections 3.1 and 3.2, respectively. Section 3.3 presents the experimental design and statistical metrics.





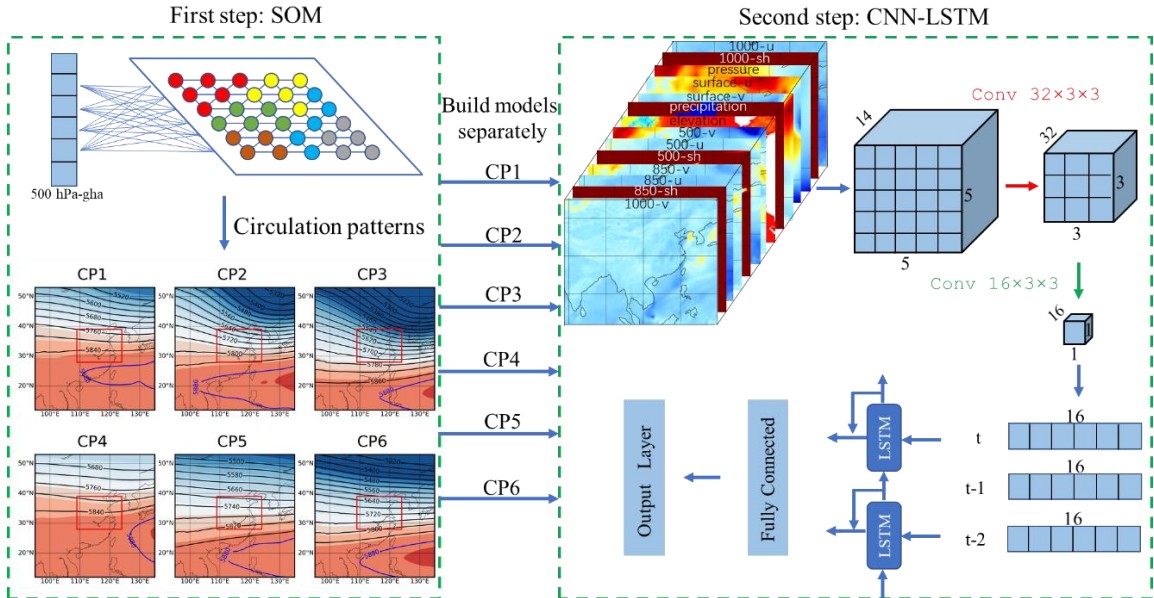

**Figure 2** The flowchart of the SOM-CNN-LSTM method. The 500 hPa-gha stands for the daily 500 hPa geopotential height anomalies; 1000-u stands for the U component of wind at 1000 hPa;1000-sh stands for the specific humidity at 1000 hPa; The pressure stands for mean sea level pressure. Others and so on

### 3.1 SOM model

The Self-organizing map (SOM) is an unsupervised neural network first introduced by Kohonen (1990) and makes no a priori assumptions about the data, which is more practical and robust than principal component analysis (PCA) or empirical orthogonal functions (EOFs) in circulation classification (Wang et al. 2019; Zhou et al. 2020). To represent daily large-scale circulation patterns over the Huaihe River basin, we use the daily 500 hPa geopotential height anomalies over the domain 95–135°E, 12–53°N as input for the SOM model. The larger domain is selected to consider the influence of multiple circulation agents on precipitation (Zhou et al. 2020). The 500 hPa geopotential height is chosen because it provides valuable information for diagnosing weather conditions in the low-level atmosphere. On the other hand, it plays a central role in controlling synoptic dynamics (Ford et al. 2015; Wang et al. 2022). The calculation formula of 500 hPa geopotential height normalized anomaly can be expressed as:

$$\langle Z \rangle = \frac{Z - Z_{mean}}{\sigma_Z} \cos \tag{1}$$

Where $Z$ is 500 hPa geopotential height, $Z_{mean}$ is the mean 500 hPa geopotential height over the quarterly average of all data, $\sigma_Z$ is the standard deviation, $\phi$ is the latitude. The cosine-latitude($\cos\phi$) is adopted to account for area differences across the grid points(Loikith et al. 2017; Mechem et al. 2018).

Choosing the optimal SOM node is also critical, so we test several SOM arrays by quantization and topological errors, including 2×2, 2×3, 2×4, 3×4 nodes, the result shows that 2×3 configuration is physically interpretable and sufficient to





capture large-scale circulation patterns. In this study, the SOM analysis is performed mainly using the Python miniSOM

library (Vettigli, 2021), and the corresponding optimal parameters are summarized in Table 1.

**Table 1** SOM optimal parameters in this study

| SOM optimal parameters | Value |
|---|---|
| Sigma | 0.5 |
| Learning_rate | 0.05 |
| Neighborhood_function | gaussian |
| Random_seed | 5 |
| Train_batch | 10000 |

**3.2 CNN-LSTM model**

CNN has the advantage of extracting distinctive spatial features from images and LSTM has the ability to deal with temporal

series data (Shen, 2018; LeCun et al. 2015). Considering that the precipitation is influenced by the surrounding topography,

the weather state of the current day and the previous days, we develop a spatiotemporal deep neural network model by

combining CNN and LSTM. We build the model in the following steps:

1. Data preparation. First, each predictor is normalized to reduce the influence of different dimensions by min-max

normalization. Second, we use the normalized data to construct input arrays with dimensions of $(508*1380)\times14\times5\times5\times3$,

where 508 represents the number of precipitation grids in the basin, 1380 represents the number of summer days, 14 is the

number of predictors, 3 represents the time dimension(i.e., t, t-1, t-2), and $5\times5$ grids are used as input to fully consider the

spatial information of each grid. Third, in order to build models separately for each circulation pattern, we divide the input

arrays into 6 groups based on the SOM results.

2. CNN model construction. Convolutional neural network (CNN) has been widely used in image recognition, object

detection, and precipitation forecasting. It can extract more abstract features from the original image through a simple

nonlinear model, avoiding the complex feature extraction process. As shown in Fig. 2, the CNN model includes an input

layer with dimensions of $14\times5\times5$, two convolutional layers, and a flattening layer. The convolution layer can extract

informative local features from the input layer, and the flattening layer converts the matrix into a one-dimensional feature

vector that is used as the input to the LSTM layer (Amini et al. 2022). Among them, the kernel size of the first convolutional

layer is set to $32\times 3 \times 3$, where 32 is the output channel number, and $3 \times 3$ is the size of the kernel. To avoid overfitting and

accelerate the training, batch normalization is applied to convolution layers (Pan et al. 2019).

3. LSTM model construction. The Recurrent Neural Network (RNN) is a kind of neural network for processing sequence

data, which can mine time series and semantic information from data. As a special RNN model, the long short-term memory

network (LSTM) can overcome the vanishing and exploding gradient problems (Hochreiter and Schmidhuber, 1997).

Besides, the interactive operation among the input gate, output gate and forget gate in LSTM enables the model to solve the

long-term dependency problem (Huang and Kuo, 2018). As shown in Fig. 2, the LSTM model includes an input layer where





the data comes from the output of the CNN, a bidirectional LSTM layer with 16 hidden units, and a fully connected layer. Considering the impact of previous meteorological information on the precipitation, the input of the LSTM model not only includes the data of the current day but also two days ago (i.e., t-1, t-2).

We select the Python package Pytorch as the framework of the above models, and the NVIDIA A5000 GPU (Graphics
Processing Unit) to accelerate model training. The hyperparameters of models, such as learning rate, epochs, and batch size, are determined by the trial-and-error method. Furthermore, the above models are trained with the Adam optimization algorithm (Kingma and Ba, 2014).

### 3.3 Experimental design and statistical metrics

To answer the three questions in the introduction, we compare the SOM-CNN-LSTM method with three other benchmark
methods including CNN, LSTM, and CNN-LSTM. The design differences of the four methods are shown in Table 2. Among them, the CNN-LSTM method is used to illustrate the effectiveness of circulation classification, while the CNN and LSTM methods are used to illustrate the importance of the incorporation of temporal and spatial information. Besides, the precipitation forecast skill continuously decreases with increasing lead times, so we build the post-processing method for each lead time separately. This means that only for the SOM-CNN-LSTM method, we need to build $15 \times 6 \times 4 = 360$ models,
where 15 represents the number of lead times, 6 represents the number of circulation patterns, and 4 represents different seasons. Therefore, to improve work efficiency, we first filter out the optimal parameter combination for one model and then adjust other model parameters based on that. In addition, each season has different training samples (Table 3) and we use four-fold cross-validation to calibrate and evaluate the model accuracy.

**Table 2** Experiment design of different methods

| Methods | Circulation patterns | Spatial information | Temporal information |
|---|---|---|---|
| SOM-CNN-LSTM | Included | Included | Included |
| CNN | Included | Included | Not included |
| LSTM | Included | Not included | Included |
| CNN-LSTM | Not included | Included | Included |

**Table 3** Training samples of different seasons

| Season | Months | Total days | Total grids | Training samples |
|---|---|---|---|---|
| Spring | Mar, Apr, May | 1380 | 508 | 701040 |
| Summer | Jun, Jul, Aug | 1380 | 508 | 701040 |
| Autumn | Sep, Oct, Nov | 1365 | 508 | 693420 |
| Winter | Dec, Jan, Feb | 1354 | 508 | 687832 |

To evaluate the performance of the post-processing results, three statistical metrics are selected, including root mean square error (RMSE), correlation coefficient (CC), and relative bias (RB).





$$RMSE = \sqrt{\frac{\sum_{i=1}^{n}(P_i - O_i)^2}{n}} \qquad (2)$$

$$CC = \frac{\sum_{i=1}^{n}(P_i - \bar{P})(O_i - \bar{O})}{\sqrt{\sum_{i=1}^{n}(P_i - \bar{P})^2} \times \sqrt{\sum_{i=1}^{n}(O_i - \bar{O})^2}} \qquad (3)$$

$$RB = \frac{\sum_{i=1}^{n}(P_i - O_i)}{\sum_{i=1}^{n} O_i} \qquad (4)$$

Where $P_i$ and $O_i$ represent simulated and observed precipitation at the $i$th point, respectively; $\bar{P}$ and $\bar{O}$ denote the average simulated and observed precipitation, respectively; $n$ is the number of samples.

## 4 Results

### 4.1 Linkages between large-scale circulation patterns and precipitation

Fig. 3 presents six large-scale circulation patterns at the lead time of 1 day in the summer of 2007-2021. It can be seen that the SOM model can well capture the key atmospheric circulation of the western Pacific subtropical high (WPSH) that affects the summer precipitation in eastern China (Zhou et al. 2020). For WPSH, pattern CP1 exceeds 30°N in the eastern zone of the Huaihe River basin, pattern CP4 extends westward to 113°E and reaches the southeast zone of the basin, while pattern CP3 is in the southeastern zone of the basin and is located around 20°N. From the perspective of geopotential height
anomalies, patterns CP2, CP3, CP5, and CP6 have similar features, with negative (positive) 500 hPa geopotential height anomalies to the north (south) of the basin, while CP1 and CP4 have positive anomalies in the entire basin.

To further characterize the relationship between circulation patterns and precipitation, we calculate the percent of each circulation pattern, the percent of rainy days, and the percent of precipitation contribution, which can be seen in Table 4. In general, CP1 and CP4 are frequent circulation patterns, and they contribute most to total summer precipitation, exceeding
40%. In contrast, CP3 has the lowest frequency (11.09 %) with a small contribution to precipitation (only 4.96 %). Besides, precipitation is more likely to occur in CP1(76.70 %) and CP4(75.86 %), although it can occur in any circulation pattern. The above results show that the change of WPSH (moving westward and expanding northward) exerts considerable impacts on precipitation in the Huaihe River Basin. On the other hand, it also indirectly confirms the effectiveness of the circulation classification.

Considering precipitation mainly occurs in summer, we only take this season as an example to analyze the results of large-scale circulation patterns and its statistical relationship with precipitation. The results of other seasons are shown in Supplement.





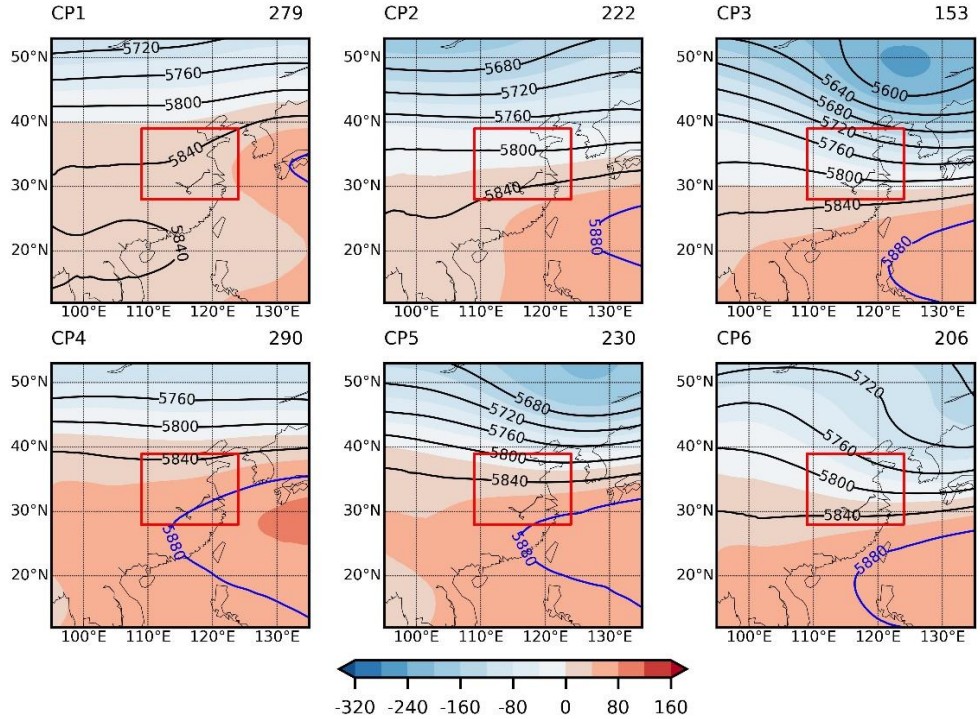

**Figure 3** Circulation patterns at the lead time of 1 day in the summer of 2007-2021. The bold blue line (5880 gpm) is the characteristic position of WPSH; The red rectangle represents the scope of the Huaihe River basin; The colored shading stands for the geopotential height anomalies at 500 hPa; The numbers for each circulation pattern are shown in the upper right corner.

**Table 4** Contribution of different circulation patterns(CPs) to summer precipitation at the lead time of 1 day during 2007-2021

| Category | CP1 | CP2 | CP3 | CP4 | CP5 | CP6 |
|---|---|---|---|---|---|---|
| CPs days | 279 | 222 | 153 | 290 | 230 | 206 |
| Precipitation days | 214 | 149 | 61 | 220 | 162 | 117 |
| Total precipitation(mm) | 1685 | 1360 | 372 | 1795 | 1334 | 957 |
| Percent of CPs days(%) | 20.22 | 16.09 | 11.09 | 21.01 | 16.67 | 14.92 |
| Percent of rainy days(%) | 76.70 | 67.12 | 39.87 | 75.86 | 70.43 | 56.80 |
| Percent of precipitation contribution(%) | 22.46 | 18.13 | 4.96 | 23.92 | 17.78 | 12.75 |





## 4.2 Overall performance of different post-processing methods


Fig. 4 shows the values of CC for different post-processing methods (i.e., SOM-CNN-LSTM, CNN, LSTM, CNN-LSTM) over 1-15 lead days during spring, summer, autumn, and winter. Overall, for each lead day and season, the four methods generally perform better than the raw forecasts. For example, the CC of the four methods ranges from 0.05 to 0.78, increased by an average of 18.69% compared with the raw forecasts. Particularly, the SOM-CNN-LSTM method performs best,
followed by CNN-LSTM, CNN, and LSTM. For instance, compared with the raw forecasts, the CC values of the SOM-CNN-LSTM method increase by an average of 32.30%, followed by 16.90%(CNN-LSTM), 13.42%(CNN), and 12.15%(LSTM).

As shown in Fig. 5, the raw forecasts have a relatively higher RMSE, Once the four post-processing methods are applied, RMSE values of the four seasons are largely decreased. Once again, the SOM-CNN-LSTM method exhibits the preferable
performance with the lowest RMSE. For example, compared with the raw forecasts, the RMSE of the SOM-CNN-LSTM method decreases by an average of 26.58%, followed by 23.64%(CNN-LSTM), 22.16%(CNN), 21.86%(LSTM).

The relative bias (RB) of the four post-processing methods is shown in Fig. 6. Similar to the above results, the SOM-CNN-LSTM method has the lowest RB. Taking summer precipitation as an example, the average RB of the SOM-CNN-LSTM method is 1.83%, CNN-LSTM is 2.12%, CNN is 2.35%, and the LSTM is 2.40%, while the average RB of the raw
forecasts is highest, reaching 2.6%, which further illustrates that the SOM-CNN-LSTM method outperforms other methods. Besides, forecast precipitation is overestimated in spring, summer and winter, and underestimated in autumn. For example, for the optimal SOM-CNN-LSTM method, precipitation is overestimated by 11.12% in spring, 1.83% in summer, 11.42% in winter, and underestimated by 4.17% in autumn. Particularly, the underestimation of the SOM-CNN-LSTM method is especially visible during the fourth lead time of summer and autumn, exceeding 15 %.
From the above results of three statistical metrics, the proposed SOM-CNN-LSTM post-processing method outperforms the no-circulation-pattern method (CNN-LSTM), the no-temporal information method (CNN), and the no-spatial information method (LSTM) at all lead times and every season, indicating that incorporating large-scale circulation patterns with local spatiotemporal information (e.g., elevation, specific humidity, and mean sea level pressure) can improve forecast skills.

We further adopt the CC and RMSE to compare the correction skills of the optimal SOM-CNN-LSTM method in different
seasons and lead times. As shown in Fig. 7(a) and 7(b), the values of CC(RMSE) continuously decrease(increase) with increasing lead times, which indicates the precipitation forecast skill has deteriorated over time. Taking 0.4 as the limit of CC, the effective lead time is 9 days in winter, 7 days in spring and autumn, and only 3 days in summer. In addition, winter forecast precipitation has the highest CC and lowest RMSE, followed by spring, autumn, and summer. The above results indicate that winter forecast precipitation performs better than other seasons, especially in summer, which is consistent with
previous studies (Buizza et al. 1999). As shown in Fig. 7(c) and 7(d), the improvement of CC(RMSE) is highest in summer with an average of 0.09(1.78), followed by 0.07(0.60) in autumn, 0.06(0.60) in spring, and 0.05(0.32) in winter, indicating that the SOM-CNN-LSTM method has better correction skills in summer. The further comparison reveals that, while the





precipitation forecast performance in winter is superior, the corrective ability is weaker. Although the summer precipitation forecast performance is not as good as the winter, it displays superior correction skills.

Since the above results show that the SOM-CNN-LSTM method has the best performance, we only use it to analyze the spatial correction skills. The first two columns in Fig. 8 show the spatial distribution of CC for the SOM-CNN-LSTM method and raw forecasts at the lead time of 1 day, revealing the significant seasonal differences in CC. For instance, for most regions of the Huaihe River basin, winter raw forecasts have the highest CC (0.55-0.75), followed by autumn (0.45-0.71), spring (0.42-0.68), and summer (0.40-0.60), and this trend remains unchanged after SOM-CNN-LSTM correction.

The third column indicates that the CC values exhibit improvement in all seasons for most regions of the basin after bias correction. Particularly, the midlands in autumn show the best correction skill (Fig. 8i), followed by summer, whereas south and northwest of the basin in spring generally show a poorer performance (Fig. 8c), and winter has the lowest improvement of CC. In addition, all seasons have relatively poor correction skills in the northwest, which may be related to the higher topography in the region.



**Figure 4** Correlation coefficient (CC) of different methods over 1-15 lead days in 4 seasons.



**Figure 5** Root mean square error (RMSE) of different methods over 1-15 lead days in 4 seasons.





**Figure 6** Relative bias (RB) of different methods over 1-15 lead days in 4 seasons.





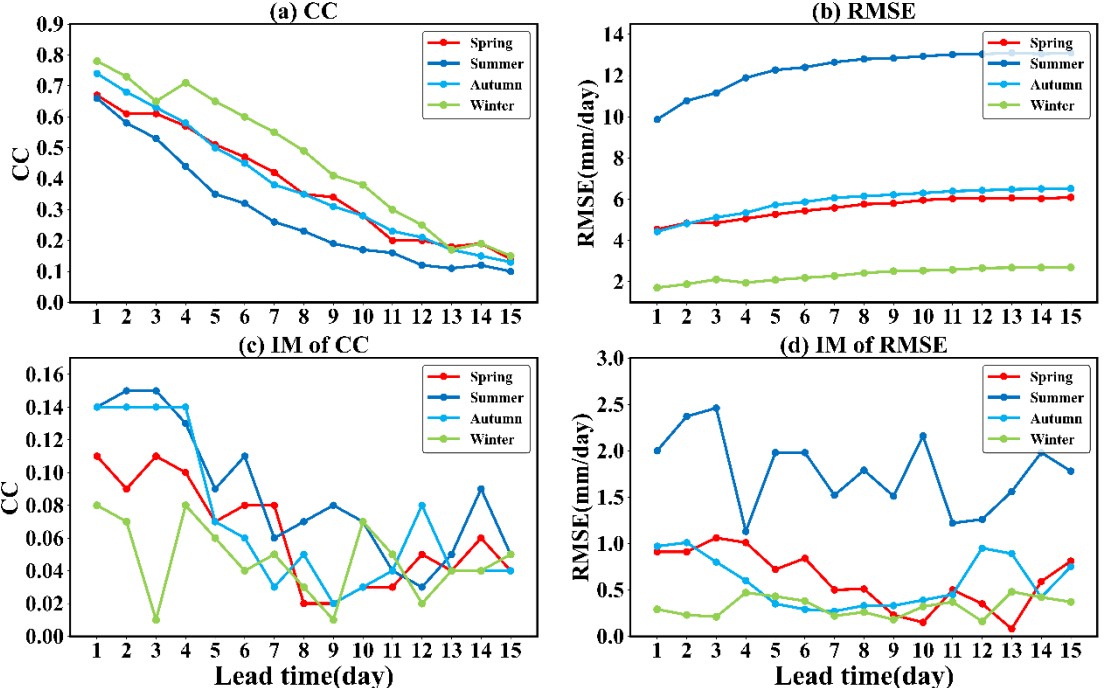

**Figure 7** (a) CC and (b) RMSE of SOM-CNN-LSTM method over 1-15 lead days during spring, summer, autumn, and winter. The second row is the (c)improvement (IM) of CC and (d) RMSE relative to raw forecasts.



**Figure 8** Spatial distributions of the CC for SOM-CNN-LSTM method and raw forecasts at the lead time of 1 day. The third column is the improvement of CC in spring, summer, autumn and winter.

## 4.3 Evaluation of inter-annual and different precipitation intensities in Summer

### 4.3.1 Inter-annual assessment of different methods

In the previous section, we mainly focus on analyzing the overall and spatial forecast skills of different post-processing methods. The forecast skills of precipitation in the time dimension may be also different. Therefore, in this subsection, we take the summer precipitation as an example to analyze the annual forecast skills of different methods. Fig. 9 presents the RB of four methods for each summer over 1-15 lead days during 2007-2021. Overall, for each year and most lead times, the




SOM-CNN-LSTM method performs best with the lowest RB. In addition, there are significant inter-annual differences in the forecast performance. For example, precipitation of all four methods is significantly underestimated for most lead times in 2013, 2018, and 2019, and overestimated in 2010, 2012, and 2015. Furthermore, when the lead time exceeds 12 days, forecast precipitation is overestimated in most years, especially in 2010 and 2014. This significant interannual difference may be related to large-scale circulation configuration.

(a) RB(%) 2007 (b) RB(%) 2008 (c) RB(%) 2009
(d) RB(%) 2010 (e) RB(%) 2011 (f) RB(%) 2012
(g) RB(%) 2013 (h) RB(%) 2014 (i) RB(%) 2015
(j) RB(%) 2016 (k) RB(%) 2017 (l) RB(%) 2018
(m) RB(%) 2019 (n) RB(%) 2020 (o) RB(%) 2021

Lead time(day)





**Figure 9** RB of different methods for each summer over 1-15 lead days from 2007 to 2021.

**4.3.2 Performance under different precipitation intensities**

We further investigate the performance of four post-processing methods at different intensities, namely 0-1, 1-5, 5-10, 20-40, and >=40 mm/d, corresponding to no rain, light rain, moderate rain, heavy rain, and violent rain, respectively (Zambrano-Bigiarini et al. 2017). Considering that precipitation mainly occurs in summer, we take the season as an example for analysis. As shown in Fig. 10, the values of RMSE for all post-processing methods are lower than raw forecasts at different

precipitation intensities, especially for no rain, light rain, and moderate rain events, which indicates that the four post-processing methods can reduce the bias and significantly improve the forecast skills. Clearly, the SOM-CNN-LSTM method achieves better scores than other methods in terms of the lowest RMSE. For example, compared with the raw forecasts, the RMSE values of the SOM-CNN-LSTM method in moderate rain events (Fig. 10c) decrease by an average of 39.70%, followed by 36.02% (CNN-LSTM), 34.95% (CNN), and 33.91% (LSTM). For heavy and violent rain events, the SOM-

CNN-LSTM method has relatively better performance under lead times ranging from 1 to 7 days, with the RMSE decreasing by 14.85% and 3.05%, respectively, whereas the advantage is no longer obvious when the lead time exceeds 7 days, the values of RMSE only decrease by 5.4% and 2.34% respectively. The reason is that the accuracy of forecast skills decreases with increasing lead times, and on the other hand, few violent rain events cannot provide enough training samples for deep learning models.



**Figure 10** RMSE of different methods over 1-15 lead days in summer at different intensities of (a) no rain, (b) light rain, (c)moderate rain, (d) heavy rain, and (e) violent rain.

## 5 Discussion

Raw precipitation forecasts usually exhibit systematic and random errors due to the initial condition, boundary condition errors, and model structural errors from NWP. Prior work has documented the effectiveness of statistical post-processing techniques in reducing these biases and improving the accuracy of NWP. For instance, Scheuerer and Hamill (2015) presented a parametric post-processing method by fitting censored, shifted gamma distributions to access the conditional



distribution of observed precipitation, which can significantly improve forecast skills. However, traditional post-processing methods generally overlook the influence of large-scale circulations and spatiotemporal information on precipitation. To

overcome the problem, we propose the SOM-CNN-LSTM post-processing method. we compare the method with other benchmarks, including CNN, LSTM, and CNN-LSTM methods. The findings of this research are as follows.

Firstly, the SOM model can well capture the westward and northward movement of the SHAP, the primary circulation system influencing summer precipitation in eastern China, suggesting the effectiveness of circulation classification using SOM. The SOM-CNN-LSTM method performs better than the CNN-LSTM method in terms of three statistical metrics,

indicating the effectiveness of considering the large-scale circulation patterns to correct the forecast precipitation. Secondly, the SOM-CNN-LSTM method performs better than CNN and LSTM methods, which indicates that considering both temporal and spatial information can improve forecast skills.

There are a growing number of deep learning models for statistical post-processing of numerical weather prediction, such as CNN (Pan et al. 2019) and ConvLSTM (Shi et al. 2015). The highlight of our work is the effective combination of the

advantages of CNN for spatial data and LSTM for time series. On the other hand, through circulation classification, the effective information of the large-scale circulation pattern (i.e., westward and northward movement of the SHAP) is subtly integrated into the deep learning model.

However, some limitations still need to be further studied. Firstly, we primarily use 500 hPa geopotential height for circulation classification, more circulation variables such as column-integrated moisture fluxes (Zhang et al. 2022), sea level

pressure (Loikith et al. 2017), and vertical velocity (Schlef et al. 2019), could also be used to represent the large-scale circulation patterns. Particularly, persistence and/or transitioning of circulation patterns may influence the local precipitation, which can be incorporated into the post-processing frame (Roller et al. 2016). Secondly, the SOM-CNN-LSTM method has relatively poor performance in heavy and violent rain when the lead time exceeds 7 days, which can be attributed to the limited violent rain samples training the model. Therefore, more studies on how to improve the forecast skills of violent rain

should be carried out. Thirdly, the spatiotemporal deep neural network can significantly improve the precipitation forecast skills, however, as a black box model, interpretability and understanding have been seen as potential weaknesses (Guidotti et al. 2019; Reichstein et al. 2019), meaning that we cannot understand how these predictors (e.g., elevation, specific humidity, and mean sea level pressure) affect the precipitation process. It will be valuable to consider interpretability in post-processing.

## 330  6 Conclusion

In this study, we propose the SOM-CNN-LSTM statistical post-processing method that combines large-scale circulation patterns with local spatiotemporal information to correct the raw ECMWF forecast precipitation over 1–15 lead days in the Huaihe River basin from 2007 to 2021. The proposed method is systematically evaluated with other benchmark methods (i.e.,



CNN, LSTM, and CNN-LSTM) in terms of root mean square error, correlation coefficient, and relative bias, and is also
evaluated from space-scale, time-scale, and intensity. The main conclusions of the study are as follows:

(1) The SOM model can effectively classify the large-scale circulation patterns over the Huaihe River basin. Particularly, the SOM can well capture the westward and northward movement of the western pacific subtropical high, and the corresponding circulation patterns CP1 and CP4 contribute the most to the total summer precipitation, exceeding 40%.

(2) The proposed SOM-CNN-LSTM post-processing method outperforms the no-circulation-pattern method (CNN-
LSTM), the no-temporal information method (CNN), and the no-spatial information method (LSTM) at all lead times and each season, and the optimal method has the largest correlation coefficient improvement (32.30%) and root mean square error reduction (26.58%). The results indicate incorporating large-scale circulation patterns with local spatiotemporal information can improve forecasting skills.

(3) There are significant seasonal and inter-annual differences in the forecast skills of precipitation. Winter precipitation
has better forecast skills than summer, whereas summer precipitation has better correction skills than winter. Summer precipitation is significantly underestimated in 2013, 2018, and 2019, and overestimated in 2010, 2012, and 2015. Furthermore, when the lead time exceeds 12 days, forecast precipitation is overestimated in most years, especially in 2010 and 2014.

(4) The SOM-CNN-LSTM method also performs best for different precipitation intensities. Particularly, for heavy and
violent rain events, the SOM-CNN-LSTM method has relatively better performance under lead times ranging from 1 to 7 days, whereas the advantage is no longer obvious when the lead time exceeds 7 days, which can be attributed to the limited precipitation samples for training the model.

In summary, this study provides a feasible and effective post-processing method to improve precipitation forecasting skills, which would benefit hydrological forecasts and other applications.


*Code and data availability*. The CN05.1 daily precipitation dataset can be obtained from Wu and Gao (2013). Forecast precipitation, meteorological factors, and 500 hPa geopotential height are from the forecast dataset of ECMWF and can be obtained from the following website: https://apps.ecmwf.int/datasets/data/tigge. Topography can be obtained from the
Geospatial Data Cloud of China (http://www.gscloud.cn/#page1/1). The code used in this study are available from the authors on request.

*Author contributions*. Zhongmin Liang: conceptualization, methodology, funding acquisition; Tuantuan Zhang: data
preparation, software, validation, visualization, writing – original draft; Wentao Li: supervision, data analysis; Jun Wang: supervision; Yiming Hu: reviewing; Binquan Li: reviewing.



*Competing interests*. The authors declare no competing financial interests.


*Financial support*. This study is supported by the National Natural Science Foundation of China (Grant No: 41730750) and National Natural Science Foundation of China (Grant No:41877147).

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
