# Peer review of "Statistical post-processing of precipitation forecasts using circulation classifications and spatiotemporal deep neural networks"

_Hydrology and Earth System Sciences, 2022_

## Author Comment (AC1)

**Response to Review**

**Reviewer #1:**

**Overall comment:**

This manuscript proposed the SOM-CNN-LSTM post-processing method to correct the raw daily forecast precipitation by combining large-scale circulation patterns with local spatiotemporal information. The proposed method showed better performance than other benchmark methods (i.e., CNN, LSTM, CNN-LSTM). The paper is very interesting, well written and well structured. We highly recommend the paper for publication with moderate revision.

We thank the reviewer for this positive evaluation and the constructive feedback. In the following, we provide a point-by-point response to the reviewers' suggestions. We are confident that all points raised can be appropriately addressed and will help to significantly improve the quality of our manuscript.

**Major comments:**

1. I think it would be good for the readers if the authors could briefly add the meaning of four-fold cross-validation in this study.

Thank you for your suggestion. We agree that it would be beneficial to the readers. For four-fold cross-validation, the 15 years of datasets are randomly grouped into four groups, and one group of datasets is selected as validation data while the other groups of datasets are used as the training data to fit the statistical post-processing models (i.e., SOM-CNN-LSTM, CNN, LSTM, and CNN-LSTM). This step will be repeated four times until all datasets are used for validation. We will add the meaning of four-fold cross-validation in the resubmitted manuscript.

2. As for predictors, why didn't the authors consider to use reanalysis data as predictors to establish the post-processing model? Based on my experience, the reanalysis data (e.g., ERA5) is more accurate than the forecast data. Meanwhile, in addition to the predictors mentioned in the paper, the vertical velocity affecting precipitation is also worth to be noted.

Thank you for your questions! For the first question, we can use the predictors(e.g., elevation, specific humidity, and mean sea level pressure) from the reanalysis data(e.g., ERA5) to train the post-processing model and get better corrections to the historical forecast data, but when we need to use the trained model to correct the future forecast precipitation, the predictors from the reanalysis data cannot be obtained and we can only use the forecast data. The difference between the two data (the accuracy of the reanalysis data is better than that of the forecast data) may make the model unstable. Therefore, we still use the predictors from the forecast data to establish the post-processing model.

For the second question, as the reviewer mentioned, the precipitation is also influenced

by the vertical velocity and we will consider using it in future research.

**3. Is the circulation pattern the same in each lead time? This point is not clear.**

We are sorry for not explaining the point clearly. We use the 500 hPa geopotential height of each lead time(e.g., 1-day, 2-day, 3-day, ..., 15-day) to classify the circulation patterns, so the circulation pattern in each lead time is different. We will explain it in the resubmitted manuscript.

**Minor issues:**

**1. Line 102. "Study area and datasets" should be "Methodology".**

Thank you for your suggestion. We will replace "Study area and datasets" by "Methodology".

**2. Line 123. "" should be "".**

Thank you for your suggestion. The other two reviewers also mentioned the same question. We will replace " $\langle Z \rangle = \frac{Z - Z_{mean}}{\sigma_Z} \cos^2 \psi$  by " $\langle Z \rangle = \frac{Z - Z_{mean}}{\sigma_Z} \cos \phi$ ".

3. Line 188-189. "southeast" and "southeastern" should be consistent.

Thank you for your suggestion. We will correct it by using 'southeast' in the sentence.

4. Line 232, "each season" is more appropriate than "every season" here.

Thank you for your suggestion. We will replace "every season" by "each season".

**5. Line 305, "we compare the method" should be "We compare the method".**

Thank you for your suggestion. We will replace "we compare the method" by "We compare the method".

We appreciate for Reviewer's warm work earnestly, and hope that the correction will meet with approval.

Once again, thank you very much for your comments and suggestions.

---

## Author Comment (AC2)

**Response to Review**

**Reviewer #2:**

The authors introduced a new statistical post-processing method by incorporating largescale circulation patterns with local spatiotemporal information, which is valuable for Hydrology and Earth System Sciences. However, it still has some questions and need a revision for publishing.

Thank you for the comprehensive and constructive review of our article. We are willing to address these comments and improve the quality of the manuscript in a revised version. Please find below some answers to your questions and explanations on how we would address your comments.

(1) Section 3 study area and datasets: The title is the same as section 2. Check the title carefully.

Thank you for your suggestion. We will replace "Study area and datasets" by "Methodology".

(2) Section 3.1 SOM model: Equation (1) may be incorrect, please check all equations to make sure all of them are correct.

Thank you for your suggestion. The other two reviewers also mentioned the same question. We will replace " $\langle Z \rangle = \frac{Z - Z_{mean}}{\sigma_Z} \cos$ " by " $\langle Z \rangle = \frac{Z - Z_{mean}}{\sigma_Z} \cos \phi$ ".

(3) Section 3.1 SOM model: How to determine the larger domain (95–135°E, 12–53°N) for circulation classification? What is the impact of watershed in China on circulation classification?

Thank you for your questions! In this study, in order to capture the circulation patterns over the Huaihe River basin as much as possible, the boundary of the circulation classification is to expand around the Huaihe River basin, so we put the Huaihe River basin in the middle of the nine-grid (3\*3, Figure 3). For the second question, the Huaihe River basin is located in the eastern part of China, using the entire Chinese basin for circulation classification may overly emphasize climatological structures over the region and fail to capture the regional variability. We prefer to use the area near the target watershed(95–135°E, 12–53°N) for circulation classification.

Figure 3 Circulation patterns at the lead time of 1 day in the summer of 2007-2021. The bold blue line (5880 gpm) is the characteristic position of WPSH; The red rectangle represents the scope of the Huaihe River basin; The colored shading stands for the geopotential height anomalies at 500 hPa; The numbers for each circulation pattern are shown in the upper right corner.

**(4) Section 3.2 CNN-LSTM model: How to consider spatial information in the CNN model? It is not clear.**

We are sorry for not explaining the point clearly. There are 508 grids in the basin and for each grid, a  $5 \times 5$  sub-grids(about 125km×125km) centered on it is extracted to fully consider the spatial information(Figure 2A). Therefore, the CNN model includes input arrays with dimensions of  $508 \times 5 \times 5$ . We will explain it in the resubmitted manuscript.

---

## Author Comment (AC3)

**Response to Review**

**Reviewer #3:**

Review on "Statistical post-processing of precipitation forecasts using circulation classifications and spatiotemporal deep neural networks"

In this manuscript, the authors have proposed a statistical post-processing method that can simultaneously take into account the effects of large-scale circulation patterns and local spatiotemporal information to calibrate the ECMWF forecast dataset for the Huaihe River basin. The study is well developed and the expected results have been achieved. The new model proposed by the authors has the best calibration capability for different seasons, lead times and precipitation intensities. Overall, the study is innovative and has a high degree of completion which deserves to be published, but some issues still need to be corrected or further clarified.

Thanks for your comprehensive review and recognition of the study contribution. The constructive comments will help us improve our manuscript after revision. We provide detailed responses to your comments and our proposed manuscript revisions in the subsequent sections.

**Major comments:**

1. In the construction of the SOM-CNN-LSTM post-processing methodology, the SOM model was used to identify and classify different large-scale circulation patterns. In selecting of the SOM node, the authors have tested that the  $2\times3$  configuration is physically interpretable. It should be explained what the node here refers to in the SOM model and what their role is. Also explain why  $2\times3$  is interpretable.

Thank you for your suggestion. The SOM nodes are the clustered large-scale circulation patterns, which need to be determined before implementing the SOM model. A fewer number of nodes in the SOM array cannot capture specific circulation patterns while a greater number of nodes would produce redundant circulation patterns that are similar. Therefore, choosing the optimal SOM node is critical. In this study, we have tested several SOM arrays by quantization and topological errors, including  $2\times2$ ,  $2\times3$ ,  $2\times4$ ,  $3\times4$  nodes, and found that 6 distinctive circulation patterns with  $2\times3$  configuration can provide enough details for physical interpretation and satisfactorily describe the variations of the synoptic situations in Huaihe River basin. For physical interpretability, there is a problem in our statement. What we want to express here is that the  $2\times3$  configuration can provide enough details for physical interpretation, because fewer nodes will yield too general patterns while too many nodes may cause its interpretation to be cumbersome or impractical. We will make these points clearer in the resubmitted manuscript.

**2. The authors used three statistical metrics in their study to evaluate the prediction skill**

and the ability of the correction, but only one of them was used to evaluate and present the results in each of the relevant experiments shown in Figures 7 to 9, respectively. Consideration could be given to including the results of all evaluation metrics from the relevant experiments in the supporting material to more fully demonstrate the features and advantages of the SOM-CNN-LSTM method.

We fully agree. Adding all evaluation metrics to the corresponding experiments will fully demonstrate the features and advantages of the SOM-CNN-LSTM method. We have made supplementary calculations on the corresponding evaluation indicators based on the original data. The main conclusions of this study have not changed and new findings will be added to the resubmitted manuscript. For example, because small deviations may lead to large relative errors (RB), winter precipitation has a larger RB although it has the highest CC and lowest RMSE compared with other seasons(Figures 7(a), 7(c), and 7(e)).

---

## Author Response (AR1)

**Reply to the comments from the editor and the reviewers**

Dear Prof. Yuan and Reviewers:

We would like to express our gratitude to you, AE and the three anonymous reviewers for your positive comments and encouragement on our research. According to the comments and suggestions, we revised the manuscript and provided a detailed point-by-point reply to all of the reviewers' comments. Besides, we also updated the supplement.

**Response to Review**

**Reviewer #1:**

Overall comment:

This manuscript proposed the SOM-CNN-LSTM post-processing method to correct the raw daily forecast precipitation by combining large-scale circulation patterns with local spatiotemporal information. The proposed method showed better performance than other benchmark methods (i.e., CNN, LSTM, CNN-LSTM). The paper is very interesting, well written and well structured. We highly recommend the paper for publication with moderate revision.

We thank the reviewer for this positive evaluation and the constructive feedback. In the following, we provide a point-by-point response to the reviewers' suggestions. We are confident that all points raised can be appropriately addressed and will help to significantly improve the quality of our manuscript.

Major comments:

1. I think it would be good for the readers if the authors could briefly add the meaning of four-fold cross-validation in this study.

Thank you for your suggestion. We agree that it would be beneficial to the readers. For four-fold cross-validation, the 15 years of datasets are randomly grouped into four groups, and one group of datasets is selected as validation data while the other groups of datasets are used as the training data to fit the statistical post-processing models (i.e., SOM-CNN-LSTM, CNN, LSTM, and CNN-LSTM). This step will be repeated four times until all datasets are used for validation. We have added the meaning of four-fold cross-validation in the resubmitted manuscript. ( Lines 183-186).

2. As for predictors, why didn't the authors consider to use reanalysis data as predictors to establish the post-processing model? Based on my experience, the reanalysis data (e.g., ERA5) is more accurate than the forecast data. Meanwhile,in addition to the predictors mentioned in the paper, the vertical velocity affecting precipitation is also worth to be noted.

Thank you for your questions! For the first question, we can use the predictors(e.g., elevation, specific humidity, and mean sea level pressure) from the reanalysis data(e.g., ERA5) to train the post-processing model and get better corrections to the historical forecast data, but when we need to use the trained model to correct the future forecast precipitation, the predictors from the reanalysis data cannot be obtained and we can only use the forecast data. The difference between the two data (the accuracy of the reanalysis data is better than that of the forecast data) may make the model unstable. Therefore, we still use the predictors from the forecast data to establish the post-processing model.

For the second question, as the reviewer mentioned, the precipitation is also influenced by the vertical velocity and we will consider using it in future research.

3. Is the circulation pattern the same in each lead time? This point is not clear.

We are sorry for not explaining the point clearly. We use the 500 hPa geopotential height of each lead time(e.g., 1-day, 2-day, 3-day, ..., 15-day) to classify the circulation patterns, so the circulation pattern in each lead time is different. We will explain it in the resubmitted manuscript. ( Line 118).

Minor issues:
1. Line 102. "Study area and datasets" should be "Methodology".

Thank you for your suggestion. We have replaced "Study area and datasets" by "Methodology". ( Line 102).

2. Line 123. "" should be "".

Thank you for your suggestion. The other two reviewers also mentioned the same question. We have replaced "$\langle Z \rangle = \frac{Z - Z_{mean}}{\sigma_Z} \cos$" by "$\langle Z \rangle = \frac{Z - Z_{mean}}{\sigma_Z} \cos\phi$". ( Line 123).

3. Line 188-189. "southeast" and "southeastern" should be consistent.

Thank you for your suggestion. We have corrected it by using 'southeast' in the sentence. ( Line 203).

4. Line 232, "each season" is more appropriate than "every season" here.

Thank you for your suggestion. We have replaced "every season" by "each season". ( Line 245).

5. Line 305, "we compare the method" should be "We compare the method".

Thank you for your suggestion. We have replaced "we compare the method" by "We

compare the method". **( Line 335).**

**Reviewer #2:**
The authors introduced a new statistical post-processing method by incorporating large-scale circulation patterns with local spatiotemporal information, which is valuable for Hydrology and Earth System Sciences. However, it still has some questions and need a revision for publishing.

Thank you for the comprehensive and constructive review of our article. We are willing to address these comments and improve the quality of the manuscript in a revised version. Please find below some answers to your questions and explanations on how we would address your comments.

(1) Section 3 study area and datasets: The title is the same as section 2. Check the title carefully.

Thank you for your suggestion. We have replaced "Study area and datasets" by "Methodology". **( Line 102).**

(2) Section 3.1 SOM model: Equation (1) may be incorrect, please check all equations to make sure all of them are correct.

Thank you for your suggestion. The other two reviewers also mentioned the same question. We have replaced "$\langle Z \rangle = \frac{Z - Z_{mean}}{\sigma_Z} \cos$" by "$\langle Z \rangle = \frac{Z - Z_{mean}}{\sigma_Z} \cos\phi$". **( Line 123).**

(3) Section 3.1 SOM model: How to determine the larger domain (95–135°E, 12–53°N) for circulation classification? What is the impact of watershed in China on circulation classification?

Thank you for your questions! In this study, in order to capture the circulation patterns over the Huaihe River basin as much as possible, the boundary of the circulation classification is to expand around the Huaihe River basin, so we put the Huaihe River basin in the middle of the nine-grid (3*3, Figure 3). For the second question, the Huaihe River basin is located in the eastern part of China, using the entire Chinese basin for circulation classification may overly emphasize climatological structures over the region and fail to capture the regional variability. We prefer to use the area near the target watershed(95–135°E, 12–53°N) for circulation classification.

[Figure]

**Figure 3** Circulation patterns at the lead time of 1 day in the summer of 2007-2021. The bold blue line (5880 gpm) is the characteristic position of WPSH; The red rectangle represents the scope of the Huaihe River basin; The colored shading stands for the geopotential height anomalies at 500 hPa; The numbers for each circulation pattern are shown in the upper right corner.

(4) Section 3.2 CNN-LSTM model: How to consider spatial information in the CNN model? It is not clear.

We are sorry for not explaining the point clearly. There are 508 grids in the basin and for each grid, a 5 × 5 sub-grids(about 125km×125km) centered on it is extracted to fully consider the spatial information(Figure S4). Therefore, the CNN model includes input arrays with dimensions of 508×5×5. We will explain it in the resubmitted manuscript. **( Lines 148-149).**

[Figure]

**Figure S4** Diagram of CNN model sub-grid data extraction

(5) Section 3.2 CNN-LSTM model: In data preparation, the authors took summer precipitation as an example for explanation, so it might be better to add "Take summer precipitation as an example" before the sentence "First, each predictor is normalized…"

We fully agree. We have added it in the resubmitted manuscript. **( Line 144).**

(6) Section 3.2 CNN-LSTM model: The authors selected 14 predictors as the input of the CNN-LSTM model and were shown in Figure 2, but it may be better to add a table for 14 predictors with corresponding description.

Thank you for your suggestion. We have added a table for 14 predictors in the resubmitted manuscript. **( Lines 171-172).**

**Table 2** The predictors in this study

| ID | Variable name | Abbreviation |
|---|---|---|
| 1 | Specific humidity(500hPa) | 500-sh |
| 2 | Specific humidity(850hPa) | 850-sh |
| 3 | Specific humidity(1000hPa) | 1000-sh |
| 4 | U component of wind(500hPa) | 500-u |
| 5 | U component of wind(850hPa) | 850-u |
| 6 | U component of wind(1000hPa) | 1000-u |
| 7 | V component of wind(500hPa) | 500-v |
| 8 | V component of wind(850hPa) | 850-v |
| 9 | V component of wind(1000hPa) | 1000-v |
| 10 | 10 metre U wind component | surface-u |
| 11 | 10 metre V wind component | surface-v |

| 12 | Surface pressure | pressure |
|----|------------------|----------|
| 13 | elevation | elevation |
| 14 | Total Precipitation | precipitation |

(7) Section 5 discussion: The following references may be helpful to discuss the violent rain.

Chen G, Wang W C. Short-Term Precipitation Prediction for Contiguous United States Using Deep Learning[J]. Geophysical Research Letters, 2022, 49(8): e2022GL097904.

Li J, Sharma A, Evans J, et al. Addressing the mischaracterization of extreme rainfall in regional climate model simulations–A synoptic pattern based bias correction approach[J]. Journal of Hydrology, 2018, 556: 901-912.

Thank you for the suggested papers. The recommended papers are very helpful to improve the quality of our manuscript. We have added them to this section in the resubmitted manuscript. ( Lines 355, 421-422, 455-458).

(8) L218: "Once the four post-processing" should be "once the four post-processing". Please check the manuscript to avoid similar errors.

We are very sorry for our incorrect writing. We have replaced "Once the four post-processing" by "once the four post-processing". (Line 231).

(9) L307 & L316: "SHAP" should be "WPSH".

We are very sorry for our incorrect writing. We have replaced "SHAP" by "WPSH". (Lines 337, 346).

(10) L320: Is "can" more accurate than "could"?

Yes, we have replaced "could" by "can". (Line 350).

**Reviewer #3:**
Review on "Statistical post-processing of precipitation forecasts using circulation classifications and spatiotemporal deep neural networks"

In this manuscript, the authors have proposed a statistical post-processing method that can simultaneously take into account the effects of large-scale circulation patterns and local spatiotemporal information to calibrate the ECMWF forecast dataset for the Huaihe River basin. The study is well developed and the expected results have been achieved. The new model proposed by the authors has the best calibration capability for different seasons, lead times and precipitation intensities. Overall, the study is

innovative and has a high degree of completion which deserves to be published, but some issues still need to be corrected or further clarified.

Thanks for your comprehensive review and recognition of the study contribution. The constructive comments will help us improve our manuscript after revision. We provide detailed responses to your comments and our proposed manuscript revisions in the subsequent sections.

Major comments:

1. In the construction of the SOM-CNN-LSTM post-processing methodology, the SOM model was used to identify and classify different large-scale circulation patterns. In selecting of the SOM node, the authors have tested that the 2×3 configuration is physically interpretable. It should be explained what the node here refers to in the SOM model and what their role is. Also explain why 2×3 is interpretable.

Thank you for your suggestion. The SOM nodes are the clustered large-scale circulation patterns, which need to be determined before implementing the SOM model. A fewer number of nodes in the SOM array cannot capture specific circulation patterns while a greater number of nodes will produce redundant circulation patterns that are similar. Therefore, choosing the optimal SOM node is critical. In this study, we have tested several SOM arrays by quantization and topological errors, including 2×2, 2×3, 2×4, 3×4 nodes, and found that 6 distinctive circulation patterns with 2×3 configuration can provide enough details for physical interpretation and satisfactorily describe the variations of the synoptic situations in Huaihe River basin. For physical interpretability, there is a problem in our statement. What we want to express here is that the 2×3 configuration can provide enough details for physical interpretation, because fewer nodes will yield too general patterns while too many nodes may cause its interpretation to be cumbersome or impractical. We will make these points clearer in the resubmitted manuscript. **(Lines 128-133).**

2. The authors used three statistical metrics in their study to evaluate the prediction skill and the ability of the correction, but only one of them was used to evaluate and present the results in each of the relevant experiments shown in Figures 7 to 9, respectively. Consideration could be given to including the results of all evaluation metrics from the relevant experiments in the supporting material to more fully demonstrate the features and advantages of the SOM-CNN-LSTM method.

We fully agree. Adding all evaluation metrics to the corresponding experiments will fully demonstrate the features and advantages of the SOM-CNN-LSTM method. We have made supplementary calculations on the corresponding evaluation indicators based on the original data. The main conclusions of this study have not changed and there are some new findings. For example, because small deviations may lead to large relative bias (RB), winter precipitation has a larger RB although it has the highest CC

and lowest RMSE compared with other seasons(Figures 7(a), 7(c), and 7(e)).

For Figures 8-10, we also add the corresponding statistical metrics and new figures have been placed in the supporting material. We also add the new findings in the resubmitted manuscript. **(Lines 252-253, 266-270, 293-299, 316-319).**

[Figure]

**Figure 7** (a) CC, (c) RMSE, and (e) RB of SOM-CNN-LSTM method over 1-15 lead days during spring, summer, autumn, and winter. The second column is the (b)improvement (IM) of CC, (d) RMSE, and (f) RB relative to raw forecasts.

[Figure]

**Figure S5** Spatial distributions of the RMSE for SOM-CNN-LSTM method and raw forecasts at the lead time of 1 day. The third column is the improvement of RMSE in spring, summer, autumn and winter.

[Figure]

**Figure S6** Spatial distributions of the RB for SOM-CNN-LSTM method and raw forecasts at the lead time of 1 day. The third column is the improvement of RB in spring, summer, autumn and winter.

[Figure]

**Figure S7** RMSE of different methods for each summer over 1-15 lead days from 2007 to 2021. The "*" indicates the best method with the lowest RMSE for each lead time.

[Figure]

**Figure S8** CC of different methods for each summer over 1-15 lead days from 2007 to 2021. The"*" indicates the best method with the highest CC for each lead time.

[Figure]

**Figure S9** RB of different methods over 1-15 lead days in summer at different intensities of (a) no rain, (b) light rain, (c)moderate rain, (d) heavy rain, and (e) violent rain.

[Figure]

**Figure S10** CC of different methods over 1-15 lead days in summer at different intensities of (a) no rain, (b) light rain, (c)moderate rain, (d) heavy rain, and (e) violent rain.

3. The study focuses on the Huaihe River basin in China. The application and development of similar research in the region should be described in the manuscript to further highlight the main purpose and innovation of this study.

We fully agree. We have found some similar research in the region. Particularly, for Huaihe River basin, Tao et al. (2014) adopted the ensemble pre-processor (EPP) method to calibrate the TIGGE multimodel ensemble forecast precipitation and Li et al. (2022b) adopted the CNN model to correct raw forecast precipitation by considering multi-spatial information. Although the above results show that post-processed precipitation forecasts have substantial improvement over the raw forecasts, these traditional post-processing methods overlook the influence of large-scale circulations and spatiotemporal information on precipitation. To overcome the problem, we propose the SOM-CNN-LSTM post-processing method. We compare the method with other

benchmarks, including CNN, LSTM, and CNN-LSTM methods. We have added these sentences to the discussion section in the resubmitted manuscript. **(Lines 328-333, 505-507).**

Reference:
Tao, Y., Duan, Q., Ye, A., Gong, W., Di, Z., Xiao, M., and Hsu, K.: An evaluation of post-processed TIGGE multimodel ensemble precipitation forecast in the Huai river basin, Journal of hydrology, 519, 2890-2905, https://doi.org/10.1016/j.jhydrol.2014.04.040, 2014.

Li, W., Pan, B., Xia, J., and Duan, Q.: Convolutional neural network-based statistical post-processing of ensemble precipitation forecasts, Journal of Hydrology, 605, https://doi.org/10.1016/j.jhydrol.2021.127301, 2022b.

Minor comments:
1. L82,305 'we' => 'We'.

We are very sorry for our incorrect writing. We have replaced "we" by "We". **(Lines 82, 335).**

2. L95 'contains' => 'contain'.

We are very sorry for our incorrect writing. We have replaced "contains" by "contain". **(Line 95).**

3. L102 The title of section 3 is wrong.

We are very sorry for our incorrect writing. We have replaced "Study area and datasets" by "Methodology". **(Line 102).**

4. L123 The formula is incomplete.

We are very sorry for our incorrect writing. We have replaced "$\langle Z \rangle = \frac{Z - Z_{mean}}{\sigma_Z} \cos$" by

"$\langle Z \rangle = \frac{Z - Z_{mean}}{\sigma_Z} \cos\phi$". **(Line 123).**

5. Change the use of color table in Figure 8. The authors use only one color table in Figure 8 to represent two types of data, correlations and changes in correlations, which can be confusing. Also, this color table is more appropriate to represent the variation between positive and negative values, which is not the case for the two variables in this figure.

Thank you for your suggestion. We have changed the figure with two color tables **(Line**

**284).** The revised figure is as follows:

[Figure]

**Figure 8** Spatial distributions of the CC for SOM-CNN-LSTM method and raw forecasts at the lead time of 1 day. The third column is the improvement of CC in spring, summer, autumn and winter.

6. In Figure 9, the conclusion the authors most wanted to express would have been the difference between the precipitation predictions for different years, but at the same time they also point out that the SOM-CNN-LSTM method performs the best. However, the color table used and the type of Figure 9 make the latter conclusion very unclear, at least compared to the other figures in the paper. Also, the correspondence between color table and value is not fixed.Therefore,the author should consider a more appropriate way of presenting the relevant conclusions.

Thank you for your suggestion. To make the latter conclusion clearer, we add the "*" to highlight the best method with the lowest RB for each lead time. In addition, we have fixed the value for all figures to better analyze the annual forecast skills of different methods. From the revised figure, we found that the underestimation is more

appropriate in 2021 than in 2013, and the overestimation is more appropriate in 2009, 2011, and 2012. Furthermore, when the lead time exceeds 12 days, forecast precipitation is overestimated in most years, especially in 2013 and 2014. We will change them in the in the resubmitted manuscript **(Lines 295-297).** The revised figure is as follows:

[Figure]

**Figure 9** RB of different methods for each summer over 1-15 lead days from 2007 to 2021. The"*" indicates the best method with the lowest RB for each lead time.

7. L307 Is the 'SHAP' used here incorrectly? If not, it is needed to clarify this abbreviation.

We are very sorry for our incorrect writing. We have replaced "SHAP" by "WPSH". **(Lines 337, 346).**

We tried our best to improve the manuscript and made some changes in the manuscript. These changes will not influence the content and framework of the paper.
We appreciate for Editors/Reviewers' warm work earnestly, and hope that the correction will meet with approval.
Once again, thank you very much for your comments and suggestions.